# MMD-associated *RNF213* SNPs encode dominant-negative alleles that globally impair ubiquitylation

Abhishek Bhardwaj[1], Robert S Banh[1,2], Wei Zhang[3], Sachdev S Sidhu[3], Benjamin G Neel[1]

**Single-nucleotide polymorphisms (SNPs) in *RNF213*, which encodes a 591-kD protein with AAA+ ATPase and RING E3 domains, are associated with a rare, autosomal dominant cerebrovascular disorder, moyamoya disease (MMD). MMD-associated SNPs primarily localize to the C-terminal region of *RNF213*, and some affect conserved residues in the RING domain. Although the autosomal dominant inheritance of MMD could most easily explained by RNF213 gain-of-function, the type of ubiquitylation catalyzed by RNF213 and the effects of MMD-associated SNPs on its E3 ligase activity have remained unclear. We found that RNF213 uses the E2-conjugating enzymes UBE2D2 and UBE2L3 to catalyze distinct ubiquitylation events. RNF213-UBED2 catalyzes K6 and, to a lesser extent, K48-dependent poly-ubiquitylation in vitro, whereas RNF213-UBE2L3 catalyzes K6-, K11-, and K48-dependent poly-ubiquitylation events. MMD-associated SNPs encode proteins with decreased E3 activity, and the most frequent MMD allele, *RNF213^{R4810K}*, is a dominant-negative mutant that decreases ubiquitylation globally. By contrast, MMD-associated *RNF213* SNPs do not affect ATPase activity. Our results suggest that decreased RNF213 E3 ligase activity is central to MMD pathogenesis.**

## Introduction

Moyamoya disease (MMD) is a rare, autosomal dominant disorder characterized by stenotic/occlusive lesions in the circle of Willis with surrounding abnormal blood vessels in the brain (Nanba et al, 2006). These vessels have a "puff of smoke" appearance in imaging studies (hence the Japanese term, "moyamoya") and are believed to develop around the occlusive lesions to compensate for lack of blood flow. MMD occurs worldwide, but its prevalence and incidence are highest in East Asians (Kim, 2016). Incidence is bimodal with peaks at ages 5–18 and 45–60, respectively (Kim, 2016). Approximately 10–40% adults and 2.8% children with MMD suffer

intracranial hemorrhage, and 50–75% of patients experience a transient ischemic attack or stroke (Scott & Smith, 2009). 15 percent of MMD cases are familial, and these have earlier mean onset (11.8 yr) compared with sporadic cases (30 yr) (Nanba et al, 2006).

MMD risk is strongly associated with single nucleotide polymorphisms (SNPs) in *RNF213* (Liu et al, 2011). On average, ~50% of Asian (and ~80% of Japanese) MMD families carry the *RNF213^{R4810K}* allele (Cecchi et al, 2014; Moteki et al, 2015). Several other rare *RNF213* variants are found in MMD patients of diverse ethnicities (Cecchi et al, 2014; Moteki et al, 2015; Kobayashi et al, 2016). Although ~2% of Japanese individuals have *RNF213^{R4810K}*, the prevalence of MMD is low (~0.006%), indicating that additional genetic and/or environmental modifiers are required for pathogenesis (Ran et al, 2013).

*RNF213* encodes an ~591-kD protein containing tandem AAA+ ATPase domains and a RING E3 domain (Fig 1A). The ATPase domains mediate RNF213 oligomerization into a homo-hexamer (Morito et al, 2014). Nucleotide (ATP/ADP) binding to the first ATPase domain stabilizes the oligomer, and destabilization occurs upon ATP hydrolysis by the second domain. RNF213 also has ubiquitin ligase activity (Liu et al, 2011), and we showed previously that its activity can affect global ubiquitylation (Banh et al, 2016). Yet although RING domains typically have E3 activity, it was reported recently that RNF213 auto-ubiquitylation using UBE2L3 as the E2 is independent of its RING domain (Ahel et al, 2020). Another recent study showed that RNF213 uses a novel "RZ" domain (Fig 1A) to ubiquitylate Lipid A of Salmonella LPS in combination with UBE2L3 (Otten et al, 2021).

E3 domains catalyze UB (ubiquitin) transfer from (an) E2 ubiquitin-conjugating enzyme(s) to lysine residues of E3-bound substrate(s), resulting in isopeptide bond formation (Deshaies & Joazeiro, 2009). Mono-ubiquitylation joins a single UB molecule to a lysine residue in the substrate, although multiple sites on the same substrate can be mono-ubiquitylated ("multi-mono-ubiquitylation"). Poly-ubiquitylation occurs when UB residues are added sequentially to a specific lysine residue in a previously conjugated UB, forming a UB chain. The N-terminal methionine and seven lysine residues (K6, K11, K27, K29, K33, K48 and K63) in UB can

[1]Laura and Isaac Perlmutter Cancer Center, New York University Langone Medical Center, New York University, New York, NY, USA   [2]Department of Medical Biophysics, University of Toronto, Toronto, Canada   [3]Banting and Best Department of Medical Research, Donnelly Centre for Cellular and Biomolecular Research, University of Toronto, Toronto, Canada

Correspondence: Benjamin.Neel@nyulangone.org
Wei Zhang's present address is Department of Molecular and Cellular Biology, College of Biological Science, University of Guelph, Guelph, Canada.

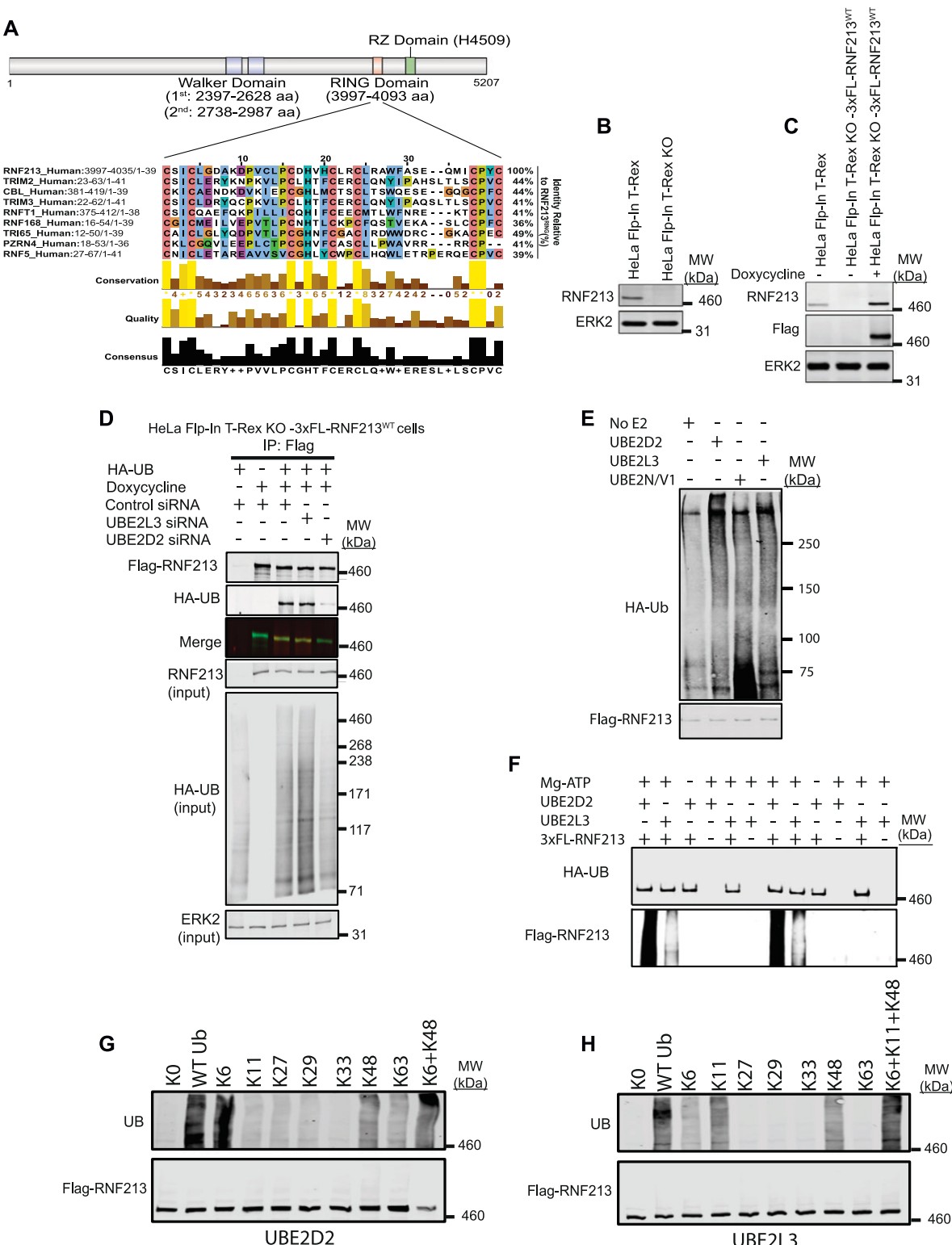

**Figure 1. RNF213 can use UBE2D2 or UBE2L3 to catalyze ubiquitylation.**
**(A)** Schematic of RNF213 structure illustrating sub-domains, and multiple sequence alignment showing closest relatives of RNF213 RING. Alignment was performed by searching the RNF213 RING sequence against the human Uniprot database, using the default settings in Blastp. The top eight sequences were selected, and an alignment diagram was generated using Jalview. **(B)** Immunoblot of single cell clone shows absence of RNF213 in HeLa Flp-In T-Rex KO cells generated by CRISPR/Cas9 technology. **(C)** HeLa Flp-In T-Rex KO cells from (B) reconstituted with doxycycline-inducible 3xFL-RNF213 were induced for 36 h, lysed, and immunoblotted for the indicated proteins. **(D)** HeLa Flp-In T-Rex KO cells expressing 3x-Flag-RNF213 after doxycycline induction were transfected with Control or *UBE2D2* or *UBE2L3* siRNAs, followed by an

participate in poly-ubiquitylation, leading to linear and/or branched UB chains (Yau & Rape, 2016). Some E2s promote specific types of UB linkages; others are more promiscuous. Most E3 ubiquitin ligases can interact with several E2s, thereby catalyzing production of different types of UB chains. Depending on the specific UB linkage, ubiquitylation regulates diverse cellular processes, including protein degradation, endocytic trafficking, inflammation, translation, and DNA repair (Yau & Rape, 2016).

Most MMD-associated SNPs map to the C-terminal region of *RNF213*, which includes the RING and RZ domains (Koizumi et al, 2016). Some occur in or near conserved RING residues (e.g., C3997Y, P4007R, D4013N, H4014N, and R4019C). However, the type(s) of ubiquitylation catalyzed by RNF213 and whether MMD-associated SNPs affect E3 ligase activity have remained unclear. Here, we report that RNF213 has K6- and K48-ubiquitin–dependent E3 ligase activity mediated by the E2 UBE2D2 and K6-, K11-, and K48-ubiquitin–dependent E3 ligase activity mediated by UBE2L3. Furthermore, MMD-associated SNPs, including the most frequent allele, $RNF213^{R4810K}$, encode proteins that decrease global ubiquitylation.

## Results

### RNF213 uses UBE2D2 or UBE2L3 to catalyze ubiquitylation

A search of the RNF213 RING against the human UniProtKB/Swiss-Prot database using Basic Local Alignment Search Tool (BLAST) (Gish & States, 1993) showed that it was most similar (44% identity) to the RING domains of TRIMs and CBL (Fig 1A). Yeast-two hybrid screens indicate that TRIMs prefer the UBE2D and UBE2E family (Napolitano et al, 2011). CBL can interact with up to 12 E2s, but uses UBE2D family E2s for the ubiquitylation of receptor tyrosine kinases, a known physiological function of CBL (Kar et al, 2012). A high proportion of other RING E3s also interact with the UBE2D class of E2s (Markson et al, 2009). An unbiased screen showed that RNF213 interacts equally well with UBE2D2 and UBE2L3 (Pao et al, 2018), whereas other work suggests that RNF213 predominantly uses UBE2L3 (Ahel et al, 2020).

We therefore evaluated the ability of these E2s to act in concert with RNF213 in a physiologic context. By using CRISPR/Cas9 technology, we generated a null (frameshift) mutant of *RNF213* in HeLa Flp-In T-Rex cells, resulting in HeLa Flp-In T-Rex KO cells (Fig 1B). We then used FLP-catalyzed recombination to generate HeLa Flp-In T-Rex KO-3xFL-RNF213^WT cells, which express wild-type (WT) 3x Flag-tagged RNF213 upon doxycycline induction (Fig 1C). Finally, Hela Flp-In T-Rex KO-3xFL-RNF213^WT cells were co-transfected with an expression construct for HA-UB (HA-tagged Ubiquitin) and siRNAs targeting *UBE2D2* or *UBE2L3* (Fig 1D). Flag-RNF213 was immunoprecipitated with anti-Flag (FL) antibody, and ubiquitinylated

RNF213 was detected by immunoblotting with anti-HA antibody. In accord with our previous findings (Banh et al, 2016), RNF213 expression (in HeLa Flp-In T-Rex KO cells) increased overall cellular ubiquitylation (compare lanes 1 and 3). *UBE2D2*-knockdown, which was validated by qRT-PCR (Fig S1A), reduced global ubiquitylation in RNF213-expressing HeLa cells to an extent comparable to that in cells lacking RNF213 expression (Fig 1D). In addition, RNF213 ubiquitylation was reduced substantially in *UBE2D2*-knockdown cells. By contrast, *UBE2L3* knockdown increased global ubiquitylation and did not affect ubiquitylation of RNF213. These results argue that at least in the context of HeLa cells, UBE2D2 (and not UBE2L3) is a(the) physiological E2 for RNF213.

Others have reported that RNF213 ubiquitylation activity in vitro is mediated by UBE2N/V1 (Takeda et al, 2020) or UBE2L3 (Ahel et al, 2020). Therefore, we tested these E2s as well as UBE2D2 for their ability to catalyze ubiquitylation by full-length immunopurified RNF213. In our hands, RNF213 auto-ubiquitylation was increased with UBE2D2 and, to a lesser extent, UBE2L3, but minimally, if at all, with UBE2N/V1 (Fig 1E). By contrast, a previous study, using different buffer conditions, showed higher levels of RNF213 auto-ubiquitylation with UBE2L3 than UBE2D2 (Ahel et al, 2020). To try to reconcile this discrepancy, we repeated our experiments using the other buffer conditions with UBE2D2 as well as UBE2L3. However, we still observed greater RNF213 auto-ubiquitylation with UBE2D2 than with UBE2L3 (Fig 1F).

Finally, we assessed the type of ubiquitin linkage added by RNF213 with UBE2D2 or UBE2L3 using UB "add-back" mutants. A UB mutant lacking all lysine residues (K0) abrogated RNF213 auto-ubiquitylation with either E2 (Fig 1G and H). Using UBE2D2, ubiquitin mutants retaining only K6 and to a lesser extent, only K48, restored RNF213 auto-ubiquitylation activity Adding K6 and K48 mutants together resulted in essentially WT levels of RNF213 ubiquitylation (Fig 1G) By contrast, with UBE2L3 ubiquitin mutants retaining only K6, only K11, or only K48 restored RNF213 auto-ubiquitylation activity to variable extents, and the three "add-back" mutants together resulted in levels comparable with WT ubiquitin (Fig 1H). Taken together, these data suggest that RNF213 can use UBE2D2 or UBE2L3 as an E2 to catalyze auto-ubiquitylation, although we find that under all conditions tested, UBE2D2 is superior. Moreover, these E2s dictate the type of ubiquitylation added. In cells (at least in HeLa cells), however, UBE2D2 appears to be the predominant E2 used by RNF213 to promote global and RNF213 ubiquitylation, although we cannot exclude the possibility that UBE2L3 has a role in other contexts.

### Moyamoya SNPs do not affect RNF213 ATPase activity

The RNF213 AAA+ ATPase domain is required for homo-hexamer formation (Morito et al, 2014). Although R4810 is located outside of

HA-UB (HA-tagged Ubiquitin) expression construct. Cells were harvested 36 h post-transfection with MG132 (10 $\mu$M) and Chloroquine (50 $\mu$M) added during the last 3 h, lysed, and immunoblotted for the indicated proteins. **(E)** HeLa Flp-In T-Rex KO cells expressing 3x Flag-RNF213^WT were treated with doxycycline for 36 h to induce Flag-RNF213^WT. Cells were harvested, and lysates were subjected to immunoprecipitation using anti-Flag antibody. Purified RNF213 was eluted using 3x Flag peptide, concentrated, and added to auto-ubiquitylation assays with UBE2D2, UBE2L3, or UBE2N/V1, as indicated. **(F)** As in (E), but with buffer conditions as from Ahel et al (2020). **(G)** Flag-RNF213^WT was purified RNF213 as in (E, F) and subjected to in vitro auto-ubiquitylation assays using UBE2D2 and the indicated ubiquitin "add-back" mutants alone or in combination. **(H)**, as in (G), but with UBE2L3 as E2.

the AAA+ ATPase and RING domains, it was reported that RNF213$^{R4810K}$ has impaired ATPase activity (Kobayashi et al, 2015). Furthermore, previous reports disagree about whether defective ATPase activity is (Kobayashi et al, 2015) or is not (Morito et al, 2014) involved in MMD pathogenesis. Both of those reports tested the ATPase activity of RNF213$^{R4810K}$. To evaluate more generally the role of RNF213 ATPase activity in MMD pathogenesis, we quantified the in vitro ATPase activity of several MMD-associated RNF213 proteins. Using FLP-catalyzed recombination, we generated HeLa Flp-In T-Rex KO lines expressing 3x-Flag-*RNF213*$^{E2488Q,E2845Q}$ (AAA+ ATPase double mutant), 3x-Flag-*RNF213*$^{I3999A}$ (RING-impaired), and four major MMD SNPs: 3x-Flag-*RNF213*$^{D4013N}$, 3x-*Flag*-*RNF213*$^{D4014N}$, 3x-Flag-*RNF213*$^{K4732T}$, and 3x-Flag-*RNF213*$^{R4810K}$ (Fig 2A). Each protein was purified by Flag-immunoprecipitation to near homogeneity (Fig 2B) and ATPase activity was assessed (Fig 2C). As expected, 3x-Flag-RNF213$^{E2488Q,E2845Q}$ had no detectable activity. The RING mutant, 3x-Flag-RNF213$^{I3999A}$, had ATPase activity comparable with that of 3x-Flag-RNF213$^{WT}$, indicating that at least in vitro, RNF213 ATPase activity is independent of its E3 ligase activity. Similarly, the proteins encoded by MMD SNPs had wild-type levels of ATPase activity, arguing that MMD is not due to defective ATPase activity, at least MMD caused by the most prevalent *RNF213* alleles.

## MMD-associated SNPs encode mutants with decreased global ubiquitylation in cells

As noted above, RNF213 forms homo-hexamers (Morito et al, 2014), so we asked whether MMD-associated alleles affect RNF213 self-association. To this end, we compared the ability of 3x-Flag-tagged (FL-)RNF213$^{WT}$, an AAA+ ATPase-inactive mutant (E2488Q), a RING E3-defective mutant (I3999A), and the most prevalent MMD variant (R4810K), to interact with EGFP-tagged WT RNF213. The AAA+ ATPase mutant alters the glutamic acid residue in the first Walker B motif (E2488), preventing ATP hydrolysis (Morito et al, 2014). As expected, FL-RNF213$^{WT}$ co-immunoprecipitated with EGFP-RNF213. Mutations in the AAA+ ATPase or RING domain did not alter binding to EGFP-RNF213; likewise, RNF213$^{R4810K}$ retained oligomerization activity (Fig 3A and B). Because FL-RNF213$^{I3999A}$ and FL-RNF213-WT associated comparably with EGFP-RNF213, E3 ligase activity is not required for, nor does it regulate, oligomerization.

Next, we performed in vitro auto-ubiquitylation assays using FL-RNF213-WT and FL-RNF213 variants purified from transiently transfected HeLa cells (Figs 3C and S2A and B). As expected, auto-ubiquitylation was observed with FL-RNF213-WT. By contrast, the MMD-associated RNF213$^{R4810K}$ mutant had lower auto-ubiquitylation activity than RNF213$^{WT}$. Previously, it was reported that transiently expressed RNF213$^{R4810K}$, but not a RING-deleted mutant, was ubiquitylated to levels similar to RNF213$^{WT}$ in 293T cells, which have low levels of RNF213 expression (Liu et al, 2011). However, our experiments show that compared with RNF213$^{WT}$, RNF213$^{R4810K}$ has decreased auto-ubiquitylation activity with UBE2D2 in vitro, although it retains a comparable activity with UBE2L3 (Fig 3C). Different *RNF213* SNPs are associated with variable degrees of MMD penetrance (Kamada et al, 2011; Liu et al, 2011; Cecchi et al, 2014; Guey et al, 2017; Sugihara et al, 2019). RNF213$^{C3997Y}$, which is encoded by an MMD allele with high

penetrance and affects a RING domain cysteine, and RNF213$^{H4014A}$, which alters a putatively critical RING domain histidine, also exhibited lower auto-ubiquitylation activity with UBE2D2 (Fig S2A and B). In concert, these results suggest that the RNF213 RING is essential for UBE2D2-, but not UBE2L3-mediated RNF213 auto-ubiquitylation.

To explore further the role of RNF213 E3 ligase activity in MMD, we assessed the E3 ligase activity associated with SNPs of different penetrance in vivo (Fig 3D and E). When reconstituted into *RNF213* KO cells, all MMD alleles tested resulted in lower global ubiquitylation than *RNF213*$^{WT}$. Intriguingly, RNF213$^{H4014N}$ and RNF213$^{C4017S}$ (encoded by high penetrance *RNF213* alleles) expression led to lower global ubiquitylation than the most prevalent, non-RING allele, *RNF213*$^{R4810K}$. Therefore, MMD penetrance might reflect the extent to which RNF213 E3 ligase activity is attenuated by a given MMD SNP in cells.

## RNF213$^{R4810K}$ acts as a dominant-negative mutant in cells

RNF213 globally affects the ubiquitylome of HER2+ breast cancer cells (Banh et al, 2016) and HeLa Flp-In T-Rex cells (Fig 1H). To assess the effect of MMD-associated RNF213 on cellular ubiquitylation, HA-tagged ubiquitin (HA-UB) was co-transfected with *RNF213* or *RNF213* mutants into (endogenous *RNF213*-replete) 293T and HeLa cells. A portion of each transfected cell population was treated with proteasomal (MG132) and lysosomal (chloroquine) inhibitors to block UB-mediated degradation. Consistent with our previous findings, over-expression of WT RNF213 resulted in an increased number of ubiquitylated proteins relative to control, vector-transfected cells (Fig 4A–D). Proteosomal and lysosomal inhibition resulted in a further increase in higher molecular weight (>150 kD) ubiquitylated species. These global changes in ubiquitylation were dependent on the RNF213 RING, as shown the effects of the RING mutant, RNF213$^{I3999A}$, which *decreased* overall ubiquitylation compared with vector-transfected cells. Expression of the AAA-ATPase mutant RNF213$^{E2488Q}$ increased overall ubiquitylation to an extent similar to that of WT RNF213, indicating that ATPase activity is dispensable for RNF213 E3 action in cells, as well as in vitro.

These results suggested that consistent with its multimeric structure, E3-defective RNF213 can act as a dominant-negative mutant. Indeed, cells over-expressing RNF213$^{R4810K}$, which also has decreased E3 ligase activity (Figs 3C and 4A–D), exhibited a global decrease in ubiquitylation compared with controls and cells over-expressing RNF213$^{WT}$ (Fig 4A–D). We also assessed the effects of *RNF213* knockdown (*RNF213*-KD). Similar to the dominant-negative effects of the RING-impaired and MMD-associated *RNF213* mutants, *RNF213*-depleted 293T and HeLa cells had a global decrease in ubiquitylation (Fig 4E and F). These observations, together with the effects of *RNF213* perturbations in breast cancer cells (Banh et al, 2016), show that RNF213 plays a crucial role in regulating ubiquitylation in multiple cell types. In breast cancer cells, RNF213-dependent ubiquitylation involved multiple proteins involved in ubiquitin metabolism, including E1s, E2s, E3s, and deubiquitylases (Banh et al, 2016). Most likely, the global increase in ubiquitylation caused by RNF213 over-expression in 293 or HeLa cells also reflects a combination of

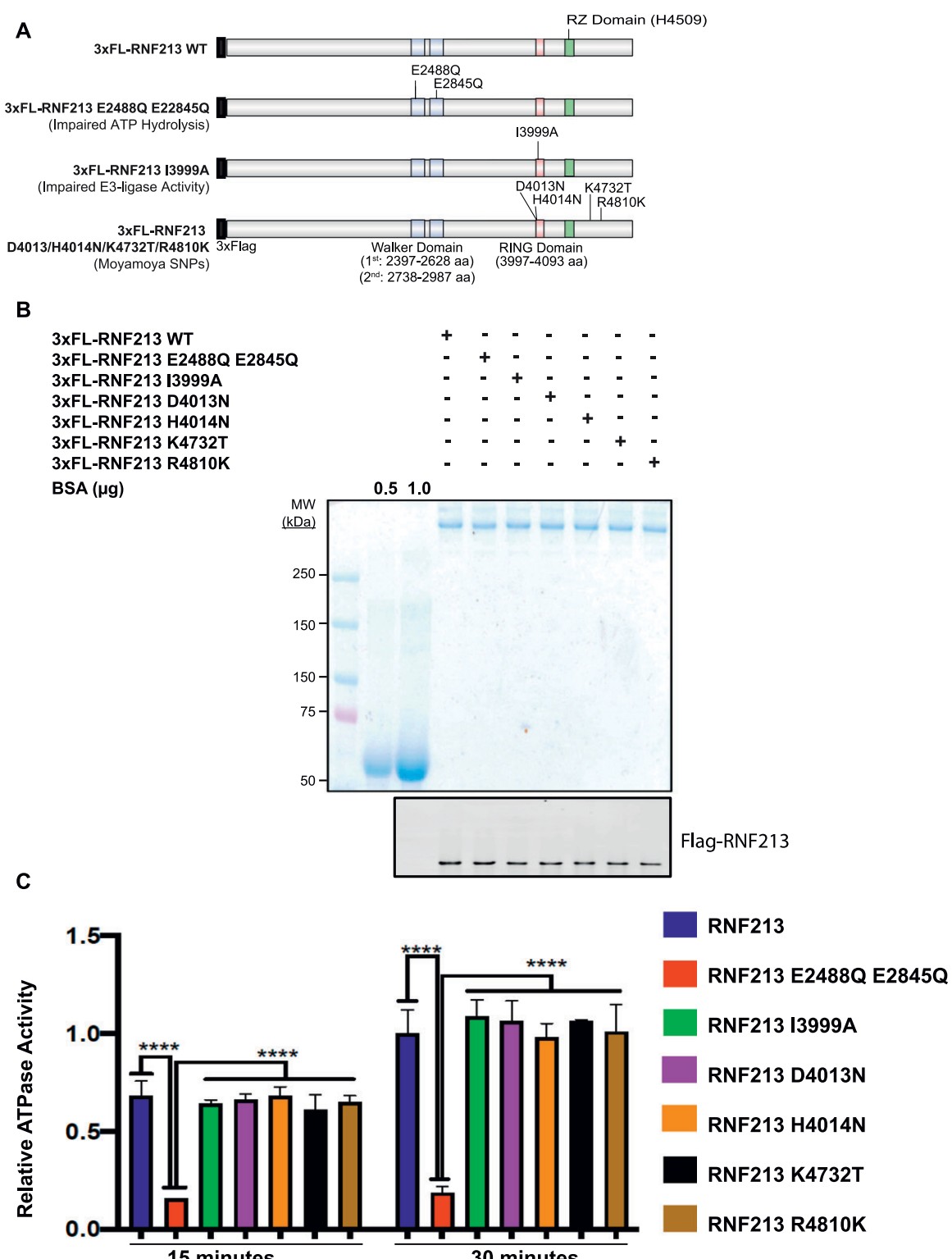

**Figure 2. MMD single nucleotide polymorphisms do not affect AAA+ ATPase activity of RNF213.**
**(A)** Schematic showing RNF213[WT] and positions of AAA+ ATPase (E2488Q, E2845Q), RING (I3999A), RZ (H4509), and/or MMD-associated (D4013N, H4014N, K4732T, and R4810K) mutants. **(B)** HeLa Flp-In T-Rex KO cells were transfected with the indicated *RNF213* expression constructs. 36 h post-transfection, RNF213 was immunopurified from cells lysates using anti-Flag antibody, resolved by SDS–PAGE and visualized by Coomassie Blue staining and immunoblotting. The indicated amounts of BSA (in µg) were loaded to allow quantification of RNF213. **(C)** ATPase activity of purified RNF213 variants from Fig 2B. Error bars represent mean ± SD of three independent experiments, each with samples in triplicate. Statistical significance was evaluated by two-way ANOVA, followed by Dunnett test.

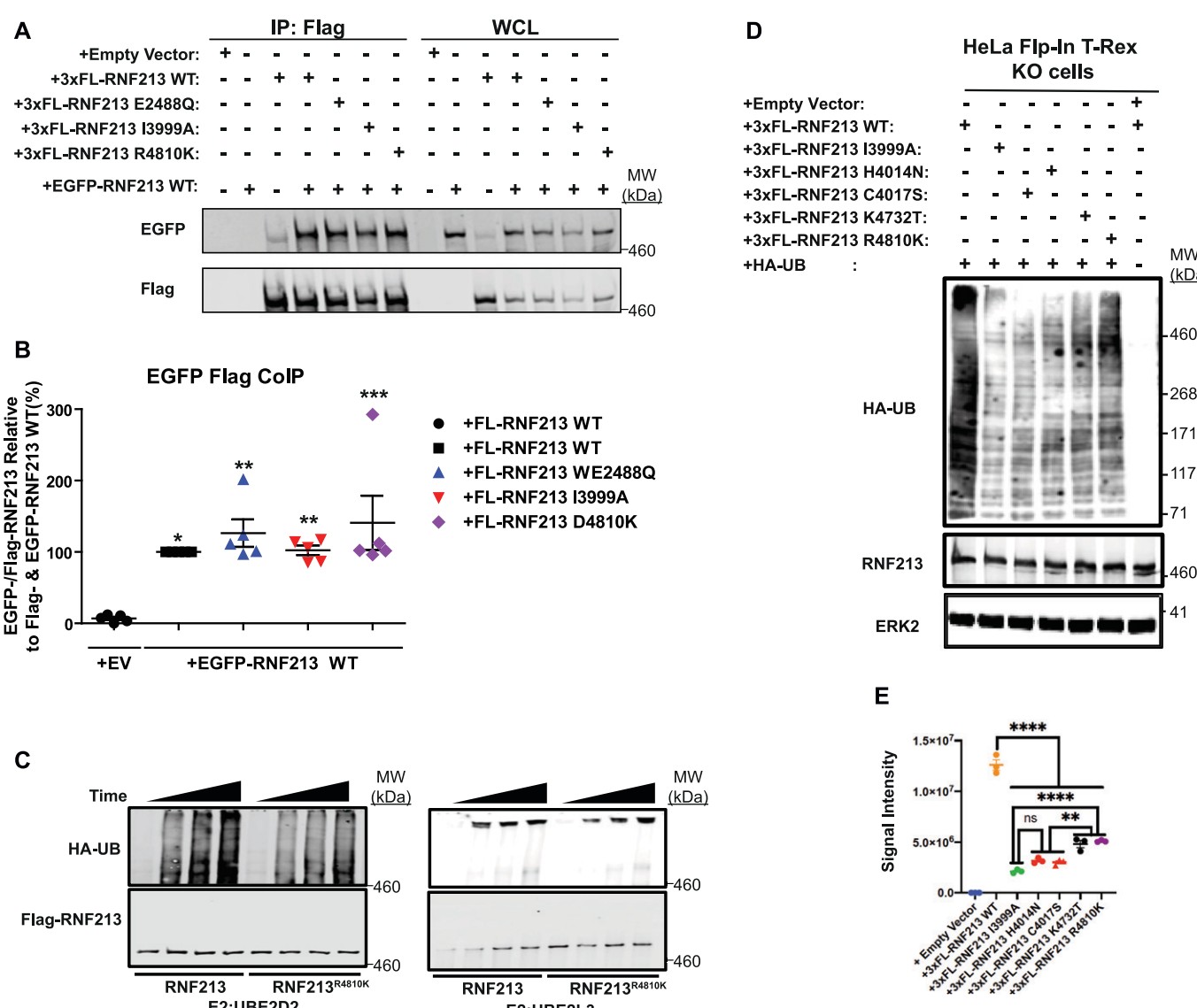

**Figure 3. MMD-associated RNF213^R4810K has decreased E3 ligase activity.**
**(A)** RNF213 mutants interact with RNF213^WT. Lysates from 293T cells co-transfected with wild-type (WT) EGFP-*RNF213*, and 3xFL-*RNF213-WT, -E2488Q, -I3999A, or -R4810K* were subjected to anti-Flag immunoprecipitation followed by anti-GFP immunoblotting. Expression of each RNF213 protein in whole cell lysates (WCL) is shown at right. **(A, B)** Quantification of levels of EGFP-RNF213 WT co-immunoprecipitated with 3xFL-RNF213^WT and RNF213 mutants in (A). (n = 5 biological replicates). Graph indicates mean percentage ± SEM. Statistical significance was evaluated by one-way ANOVA, followed by Bonferroni post hoc test. **(C)** in vitro auto-ubiquitylation assays of anti-Flag immunoprecipitates from lysates of HeLa Flp-In T-Rex KO cells expressing 3xFL-RNF213^WT or RNF213^R4810K with UBE2D2 (left) or UBE2L3 (right). **(D)** HeLa Flp-In T-Rex KO cells expressing 3xFL-RNF213^WT or the indicated RNF213 variants were transfected with an HA-UB expression construct. Cells were harvested 36 h post-transfection, and immunoblotting was performed for the indicated proteins. Some cells (as indicated) were subjected to MG132 and chloroquine for the last 3 h before harvesting. **(E)** Quantification of immunoblots from (E) (n = 3 biological replicates), showing the effects of RNF213 variants encoded by MMD single-nucleotide polymorphisms with high penetrance (red circles), low penetrance (purple circle) and undetermined penetrance (black circles), compared with a mutant in an essential RING domain cysteine, –C4017S (red triangle). Graph indicates integrated signal intensity ± SEM. Statistical significance was evaluated by one-way ANOVA, followed by Tukey test.

direct and indirect effects on the cellular ubiquitylation machinery.

# Discussion

RNF213 contains AAA-ATPase and RING E3 ligase domains in the same molecule, is associated with MMD (Liu et al, 2011), and is required for the toxic effects of PTP1B deficiency/inhibition in

hypoxic HER2+ breast cancer cells (Banh et al, 2016). Previous studies showed that RNF213 oligomerizes via its ATPase domain (Morito et al, 2014) and has ubiquitin ligase activity (Kobayashi et al, 2015; Banh et al, 2016). ATPase activity also has been implicated in recruitment of RNF213 to, and stabilization of, lipid droplets (Sugihara et al, 2019). However, the type of UB linkages added by RNF213, the cooperating E2(s), and the role(s) of the AAA-ATPase and E3 domains in MMD pathogenesis has remained unknown/controversial. We find that RNF213 can use UBE2D2 (K6>>K48) or

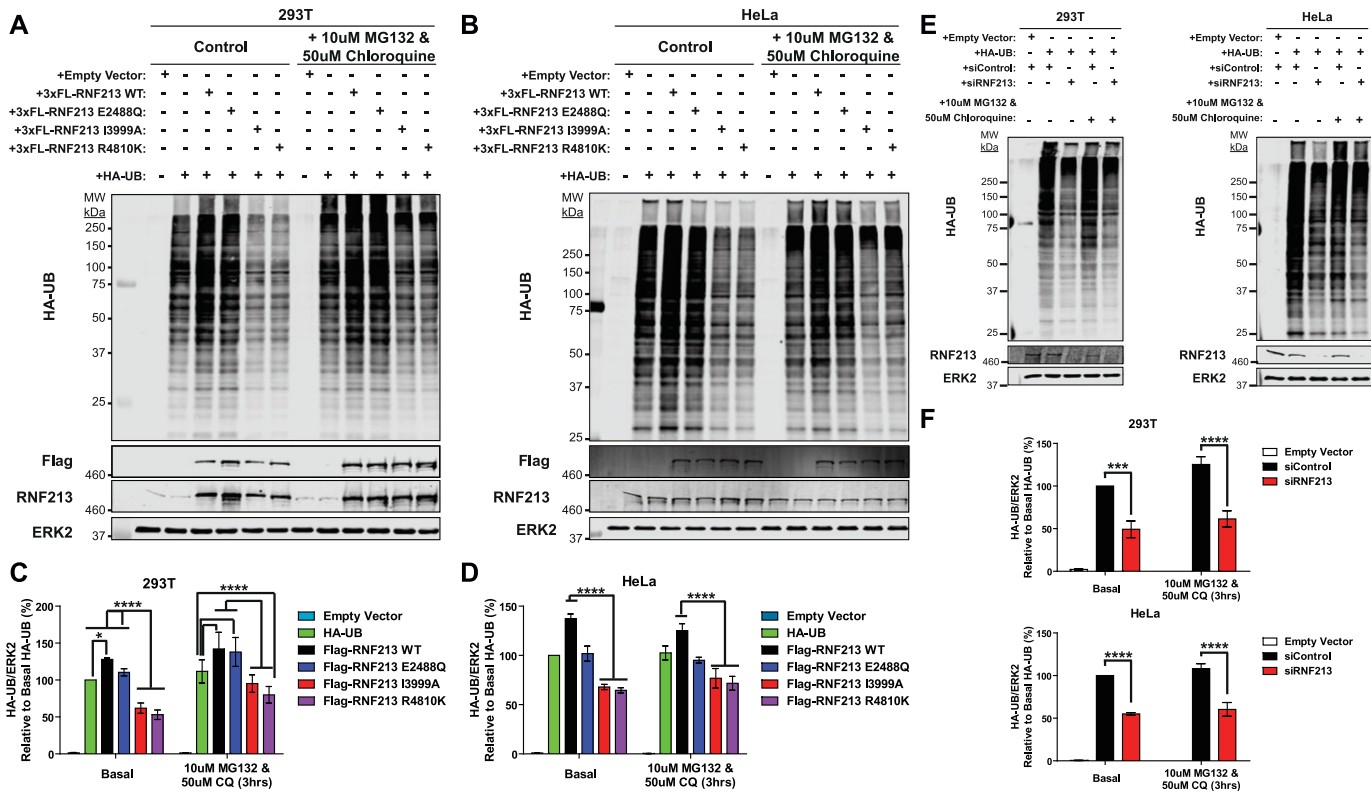

**Figure 4. E3 ligase–impaired RNF213 acts as a dominant-negative mutant that affects global ubiquitylation.**
**(A, B)** Immunoblots for HA-UB from 293T (A) and HeLa (B) cells transiently co-transfected with HA-UB and 3xFlag-*RNF213*[WT] or the indicated *RNF213* mutant expression constructs. A portion of each transfected cell population was treated with the indicated concentrations of proteasomal (MG132) and lysosomal (chloroquine) inhibitors for 3 h. **(C, D)** quantification of immunoblots from (A) and (B), respectively (*n* = 5 and 6 biological replicates for HeLa and 293T cells, respectively). **(E)** 293T and HeLa cells were transfected with Control or *RNF213* siRNAs, followed by an HA-UB expression construct, and treated with the indicated proteasome and lysosome inhibitors for 3 h. **(F)** Lysates were subjected to anti-HA immunoblotting, and results are quantified in (F) (*n* = 4 biological replicates). ERK2 serves as a loading control. Graphs indicate mean percentage of cells ± SEM. Statistical significance was evaluated by two-way ANOVA, followed by Bonferroni post hoc test.

UBE2L3 (K11, K48>K6) to catalyze distinct ubiquitylation events in vitro. We failed to observe significant RNF213-catalyzed ubiquitylation with UBE2N/UBE2V1, in contrast to a previous report that RNF213/UBE2N/UBE2V1 generates K63 ubiquitin linkages (Takeda et al, 2020). In HeLa cells, however, our knockdown experiments indicate that RNF213 ubiquitylation and RNF213-evoked global ubiquitylation requires UBE2D2. Furthermore, whereas RNF213 variants encoded by MMD alleles have apparently unaltered ATPase activity, their E3 ligase is impaired, and the extent of impairment is greater in more penetrant alleles, such as those affecting the RNF213 RING domain. Consistent with its oligomeric structure, MMD alleles act as dominant-negative alleles in cells, providing a potential explanation for the autosomal dominant inheritance pattern of this syndrome.

The finding that mutations affecting the RING domain impair in vitro ubiquitylation with UBE2D2, have more profoundly decreased RNF213-evoked global ubiquitylation in cells, and have more potent dominant-negative effects argue for a key role of RING-catalyzed ubiquitylation events in MMD pathogenesis. We reported previously that that PTP1B deficiency or inhibition increases RNF213 E3 ligase activity (Banh et al, 2016). In that study, di-Gly antibody-based UB proteomics revealed increased K6/11, K29, and K33 ubiquitylation in PTP1B-deficient cells (Banh et al, 2016). Consistent with our new

findings, those increases were normalized by *RNF213* knockdown. By contrast, an elegant recent study implicated the RNF213 RZ domain in LPS ubiquitylation (Otten et al, 2021). Notably, RZ domain-dependent ubiquitylation appears to use UBE2L3 as the E2 (Otten et al, 2021), which could explain why we failed to see a significant decrease in RNF213-evoked global protein ubiquitylation upon *UBE2L3* knockdown. Taken together, these results raise the exciting possibility that, by using different E2s, RNF213 acts as an E3 ligase both for lipids and proteins. Such dual substrate specificity is reminiscent of the tumor suppressor gene PTEN, which acts as both a lipid- and protein-phosphatase (Maehama & Dixon, 1998; Myers et al, 1998).

With UBE2D2, RNF213 appears to predominantly catalyze K6 ubiquitylation. Only three other ubiquitin ligases in mammalian cells are known to mediate K6 ubiquitylation: PARKIN, HUWE1, and BRCA1:BARD1 heterodimer (Wu et al, 2008; Yau & Rape, 2016; Michel et al, 2017). K6-UB linkages contribute to PARKIN-dependent mitophagy (Yau & Rape, 2016), whereas HUWE1 regulates K6-UB linkages on mitofusin-2 (Michel et al, 2017). BRCA1:BARD1, which assembles K6-linkages on itself and its substrates (nucleophosmin, CTIP (C-terminal binding protein 1), and RNA polymerase subunit RPB8), plays a key role in DNA replication and homologous recombination repair. K6-poly-ubiquitylated BRCA1 is recognized,

recruited, and stabilized at sites of DNA damage by RAP80 (receptor-associated protein 80). BRCA1 K6-linkages also recruit UBXN1 (UBX domain-containing protein 1), which binds to and acts as a cofactor for p97 VCP (valosin-containing protein) to negatively regulate BRCA1:BARD1 E3 ligase activity (Wu et al, 2008). Intriguingly, p97 VCP is an AAA-ATPase (Meyer et al, 2012). Notably, ATPase and E3 ligase activity reside in a single polypeptide in RNF213, whereas these functions are divided among several species in the p97 VCP/BRCA1:BARD1 interaction. PARKIN, HUWE1, and BRCA1:BARD1 also can promote other ubiquitin linkages to form heterotypic chains.

We also find that MMD-associated SNPs impair RNF213 E3 ligase activity but not ATPase activity. Furthermore, our results suggest that the penetrance of different SNPs might reflect the degree to which the allele impairs E3 ligase activity, although analysis of larger panels of *RNF213* alleles will be required to solidify these findings. A recent study suggested that RNF213 localizes to and stabilizes lipid droplets by removing the lipase ATGL (Sugihara et al, 2019). Intriguingly, ATPase and E3 ligase activity both were important for stable association with lipid droplets, although E3 activity was dispensable for ATGL expulsion. Moreover, MMD SNPs with high penetrance (C3997Y, H4014N, C4017S, and C4032R) had more impaired lipid droplet targeting/droplet stabilization than MMD SNPs with low penetrance (D4013N, R4810K). It has also been recently reported that RNF213 acts as a ISG15 sensor on lipid droplets upon interferon signaling (Thery et al, 2021). Our finding that MMD penetrance correlates with the degree of impairment of RNF213 E3 ligase activity comports with the notion that aberrant regulation of RNF213 association with lipid droplets could be central to MMD pathogenesis.

At first glance, the finding that RNF213$^{R4810K}$ has impaired E3 ligase activity would seem to be inconsistent with the autosomal dominant inheritance of MMD. However, we find that this MMD variant, as well as an engineered RING domain mutant, has dominant-negative effects. Like RNF213$^{WT}$, RNF213$^{R4810K}$ and RNF213$^{I3999A}$ retain the ability to oligomerize (Morito et al, 2014), indicating that decreased auto-ubiquitylation activity cannot be explained by inability to form a homo-hexamers. E3 activity also does not depend on RNF213 AAA+ ATPase activity. Taken together, our findings provide a biochemical explanation for the observed dominant-negative effects of E3-defective RNF213 mutants in cells. As in other multimeric complexes, such as p97 VCP and RVB1/2 (RuvB family ATP-dependent DNA helicase pontin), incorporation of one or a few defective subunits to the RNF213 holoenzyme might disable the entire complex (Jha & Dutta, 2009; Tresse et al, 2010). However, it remains unclear how MMD SNPs that map outside of the RING and AAA+ ATPase domains (e.g., RNF213$^{R4810K}$), impair the E3 activity of other complex members.

Given that our results implicate defective RNF213 activity in MMD pathogenesis, it is surprising that MMD-like phenotypes are absent in *Rnf213*$^{-/-}$ or *Rnf213*$^{R4828K/+}$ mice. Furthermore, ~2% of the Japanese population have *RNF213* SNPs, yet the incidence of MMD in Japan is as low as 0.53 per 100,000. Notably, incidence is higher in children and adolescents (5–14 yr old) and is ~2-fold more common in females than in males. MMD also is associated with anemia, autoimmunity, infection/inflammation, and radiation. Together, these findings predict that genetic or environmental modifier(s) are required to trigger MMD in patients bearing *RNF213* SNPs (and

presumably in *Rnf213*$^{-/-}$ or *Rnf213*$^{R4828K/+}$ mice). It will be important to identify such modifiers and determine how they affect RNF213 E3 activity or its substrates. A clue might come from considering biological differences between mice and fish, as *Rnf213* morphants in zebrafish do show abnormal vascular growth (Liu et al, 2011).

## Materials and Methods

### Cell lines and culture conditions

HeLa Flp-In T-Rex, HeLa, and 293T cells were grown in DMEM with 10% FBS (Invitrogen), 100 U/ml penicillin, and 100 mg/ml streptomycin at 37°C. HeLa Flp-In T-Rex were purchased from Thermo Fisher Scientific. HeLa and 293T cell lines were obtained from the ATCC, authenticated by STR testing, and assessed monthly for the absence of mycoplasma contamination (MycoAlert, Lonza). Transfections were carried out by using Fugene 6 (Promega), according to the manufacturer's instructions. For transient transfection experiments, the amount of plasmid DNA was kept constant between conditions in every experiment by adding appropriate amounts of empty vector. Control siRNAs (Cat. no. D-001810-10-05) or siRNAs targeting *RNF213* (Cat. no. L-023324-00-0005), *UBE2D2* (Cat. no. L-010383-00-0005), or *UBE2L3* (Cat. no. L-010384-00-0005) (Dharmacon), were introduced into cells using Lipofectamine RNAiMAX (Thermo Fisher Scientific), as per the manufacturer's instructions.

*RNF213*-knockout HeLa Flp-In T-Rex cells were generated using CRISPR/Cas9 technology (Ran et al, 2013). Briefly, an sgRNA targeting the third exon of *RNF213* (ACAATGGCGTCGGCCTCGGA) was designed by using http://crispr.mit.edu and cloned into the BbsI site of pSpCas9 (BB)–2A–Puro (PX459; Addgene). HeLa cells were transiently transfected with the PX459–sg*RNF213*-exon3 vector using FuGENE 6 Transfection Reagent (Promega). 48 h post-transfection, cells were diluted and plated in 96 well plate (1 cell/well). *RNF213*-knockout clones were identified by DNA sequencing and confirmed by immunoblotting. For generating isogenic HeLa Flp-In T-Rex knockout cells expressing WT RNF213 and RNF213 variants upon doxycycline induction, pcDNA5/FRT/TO-based constructs expressing 3x-Flag tagged RNF213 or RNF variants were co-transfected with pOG44, which directs constitutive expression of Flp recombinase. Cells with successful integration of 3x-Flag-*RNF213* or its variants were selected by adding hygromycin B (200 μg/ml) at 48 h post-transfection. RNF213 expression was induced by adding 1 μg/ml doxycycline.

### Immunoblotting and immunoprecipitation

Cells were lysed in lysis buffer (50 mM Tris·HCl [pH 8], 150 mM NaCl, 2 mM EDTA, 10 mM Na4P2O7, 100 mM NaF, 2 mM Na3VO4, 1% [vol/vol] NP-40, 40 μg/ml phenylmethyl sulfonyl fluoride, 2 μg/ml antipain, 2 μg/ml pepstatin A, 20 μg/ml leupeptin, and 20 μg/ml aprotinin). For whole cell lysate immunoblots, equal amounts of protein per sample were subjected to SDS–PAGE and transferred to a polyvinylidene fluoride (PVDF) membrane (Millipore). For immunoprecipitations, lysates were incubated with appropriate

antibody-agarose beads, as indicated, for 4 h on a rotator-mixer at 4°C. Beads were washed five times in lysis buffer and then incubated in sample buffer. Lysates and immunoprecipitates were resolved by modified SDS–PAGE on 3–8% Tris-acetate gels or 5% Tris-glycine gels. Gels were transferred in 1× transfer buffer, 2% methanol for 16 h at 25 V. The following antibodies were used: RNF213 (Cat. no. ABC1391; Millipore), HA (Cat. no. 3724; Cell Signaling Technology), GST (Cat. no. sc-138; Santa Cruz), Flag (Cat. no. F1804; Sigma-Aldrich), ERK2 (Cat. no. sc-1647; Santa Cruz), and GFP (Cat. no. 2956; Cell Signaling Technology). Blots were quantified by acquisition software, Image Studio Lite, using an Odyssey Infrared imaging system (Li-Cor Biosciences). Effects on overall ubiquitylation in Figs 3D and E and 4A–F were assessed by integrating the intensity of the HA-UB signal in each lane.

### In vitro ubiquitylation assays

Full-length RNF213 was purified from HeLa Flp-In T-Rex KO expressing 3x-Flag tagged wild-type *RNF213* or *RNF213* variants. RNF213 was induced with doxycycline (1 $\mu$g/ml) and lysed in RIPA buffer. Lysates were subjected to immunoprecipitations with anti-FLAG M2 magnetic beads (Cat. no. M8823; Sigma-Aldrich) for 3 h on a rotator-mixer at 4°C. Beads were collected at bottom of the tube using a magnetic stand, supernatants were discarded, and immunoprecipitants were washed five times in lysis buffer followed by elution with 50 $\mu$l of 100 ng/$\mu$l 3x-Flag peptide. Briefly, 0.1 $\mu$M of purified RNF213 was used for in vitro auto-ubiquitylation reaction in the presence of 50 $\mu$M HA-Ub (Boston Biochem), 100 nM E1 (Boston Biochem), 1 $\mu$M E2 (Boston Biochem), 10 mM Mg-ATP (Boston Biochem) for the times indicated. For Ubiquitin linkage analysis, a 2-h incubation and ~0.25 $\mu$M of purified RNF213 was used in the presence of 100 $\mu$M Ub and its linkage-specific addback mutants (Boston Biochem), 100 nM E1 (Boston Biochem), 1 $\mu$M E2 (Boston Biochem), and 10 mM Mg-ATP (Boston Biochem). Total volume of the reaction was 25 $\mu$l. Reactions were terminated by adding SDS–PAGE sample buffer, and ubiquitin incorporation was analyzed by immunoblotting, as described above.

### Plasmids

*RNF213* mutants were generated using by PCR or synthesized by Genewiz. Fragments containing the desired mutation were sequenced and cloned into wild-type *RNF213* plasmids using Gibson assembly (Gibson et al, 2009).

### Primers used for sequencing

E2488Q: *CGCCAATAAGGACCAACATCAGTTG*
E2845Q: *AGTGCGCCCGCTTTCAGCAG*
For RING mutants: *ATGCAAGGAGACAGCCAGCA*
K4732T and R4810K: *GCGTATCAGCTCTAACCCTG*

### ATPase assays

ATPase assays using full-length RNF213 were carried out using the ATPase Activity Assay Kit (Cat. no. K417-100; BioVision) as per the manufacturer's instructions. Briefly, purified RNF213 (3x-Flag-RNF213$^{WT}$, 3x-Flag-RNF213$^{E2488Q,E2845Q}$, 3x-Flag-RNF213$^{I3999A}$, 3x-Flag-RNF213$^{C3997Y}$, 3x-Flag-RNF213$^{D4013N}$, 3x-Flag-RNF213$^{H4014N}$, or 3x-Flag-RNF213$^{R4810K}$) was incubated in assay buffer containing ATPase substrate, developed at 25°C for 15–30 min, and A$_{650}$ was measured using FlexStation 3 Multi-Mode Microplate Reader. For full-length RNF213 purification, HeLa Flp-In T-Rex KO expressing 3x-Flag tagged wild-type RNF213 or RNF213 variants were induced with doxycycline (1 $\mu$g/ml) and lysed in ice cold ATPase Assay Buffer provided with the ATPase assay kit. Lysates were subjected to immunoprecipitations with anti-FLAG M2 magnetic beads (Cat. no. M8823; Sigma-Aldrich) for 3 h on a rotator-mixer at 4°C. Beads were collected at bottom of the tube using a magnetic stand, supernatants were discarded, and immunoprecipitates were washed five times in lysis buffer followed by elution with 50 $\mu$l of 100 ng/$\mu$l 3x-Flag peptide.

### Ubiquitylome analysis

Where indicated, cells were transfected with empty pRK5 vector or pRK5 expressing HA-UB (HA-tagged ubiquitin) by using Fugene 6 (Promega), as per the manufacturer's instructions. In some experiments, a portion of the transfected cells was treated with 10 $\mu$M MG132 (Tocris) and 50 $\mu$M chloroquine (Tocris). Cells were lysed in lysis buffer (50 mM Tris·HCl [pH 8], 150 mM NaCl, 2 mM EDTA, 10 mM Na$_4$P$_2$O$_7$, 100 mM NaF, 2 mM Na$_3$VO$_4$, 1% [vol/vol] NP-40, 40 $\mu$g/ml phenylmethyl sulfonyl fluoride, 2 $\mu$g/ml antipain, 2 $\mu$g/ml pepstatin A, 20 $\mu$g/ml leupeptin, and 20 $\mu$g/ml aprotinin) along with deubiquitylase inhibitor (+10 mM IAM and NEM), and lysates were analyzed by immunoblotting as described above.

### Quantitative real-time PCR

Total RNA was extracted using miRNeasy kit (Cat. no. 217004; QIAGEN). cDNA synthesis was carried out using the RT2 first stand kit (QIAGEN) cDNA synthesis kit following the manufacturer's protocol. qPCR then was carried out using Maxima SYBR Green Supermix (Life Technologies) in an Applied Biosystems StepOne Real-time PCR machine. 18S rRNA was used as the internal control for all samples. For human *UBE2D2*, the following primers were used: 5′-*GGTCACAGTGGTCTCCAGCACTAA*-3′ and 5′-*ACTTCTGAGTCCATTCCCGAGCT*-3′. For human *UBE2L3*, the following primers were used: 5′-*CATCGATGAGAAGGGGCAG* and 5′-*ACTTGGTCAGTCTTGGTGGC*-3′.

### Statistical analysis

Sample sizes and statistical tests for each experiment are mentioned in the figure legends. All analyses and graphs were generated using GraphPad Prism 9.

# Supplementary Information

# Acknowledgements

This work was funded by R01 CA49152 to BG Neel. RS Banh was supported by a Doctoral Fellowship Grant from the Canadian Breast Cancer Foundation. W Zhang was supported by a Canadian Institutes of Health Research Post-doctoral Fellowship Grant.

## Author Contributions

A Bhardwaj: resources, data curation, formal analysis, investigation, methodology, and writing—original draft, review, and editing.
RS Banh: data curation, formal analysis, investigation, and writing—original draft.
W Zhang: resources and investigation.
SS Sidhu: conceptualization and supervision.
BG Neel: conceptualization, formal analysis, supervision, funding acquisition, project administration, and writing—original draft, review, and editing.

## Conflict of Interest Statement

BG Neel is a co-founder, has equity in, and receives consulting revenue from Northern Biologics, Navire Pharmaceuticals, and Jengu Therapeutics. He is a member of the SAB, holds equity in, and receives consulting fees from Arvinas, Inc and a member of the SAB and holds equity in Recursion Pharmaceuticals. He is an expert witness for Johnson and Johnson in the ovarian cancer talc litigation in US Federal Court and has consulted for MPM Capital, Gerson Lehman Group, and Halda, Inc. None of these interests is directly relevant to the work herein.

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
