## [Reviewer comments · Life Science Alliance]

Life Science Alliance

MMD-Associated RNF213 SNPs Encode Dominant Negative Alleles That Globally Impair Ubiquitylation

Abhishek Bhardwaj, Robert Banh, Wei Zhang, Sachdev Sidhu, and Benjamin Neel

DOI: <https://doi.org/10.26508/lsa.202000807>

Corresponding author(s): Benjamin Neel, New York University Langone Medical Center

Review Timeline:

Submission Date:	2020-06-02
Editorial Decision:	2020-07-14
Revision Received:	2021-12-13
Editorial Decision:	2022-01-17
Revision Received:	2022-01-27
Accepted:	2022-01-28

Transaction Report:

July 14, 2020

Re: Life Science Alliance manuscript #LSA-2020-00807-T

Dr. Benjamin G. Neel
New York University Langone Medical Center
Laura and Isaac Perlmutter Cancer Center
522 First Avenue
New York 10016

Dear Dr. Neel,

Thank you for submitting your manuscript entitled "Moyamoya Disease-Associated RNF213 Alleles Encode Dominant Negative Alleles That Globally Impair Ubiquitylation" to Life Science Alliance. The manuscript was assessed by expert reviewers, whose comments are appended to this letter.

We share the referees enthusiasm for your dataset and would warmly welcome the submission of a revised manuscript for publication in LSA.

Please include in a revision all of the experiments suggested by the reviewers, particularly the control experiments. Please also include validation for the assay to determine the E3-ligase of RNF213 (ref #1), noting ref #2's point about other chain linkages. If this is feasible, we would suggest also to add experiments under more physiological conditions (ref #1), but this is not a precondition for re-submission.

In our view these revisions should typically be achievable in around 3 months. However, we are aware that many laboratories cannot function fully during the current COVID-19/SARS-CoV-2 pandemic and therefore encourage you to take the time necessary to revise the manuscript to the extent requested above. We will extend our 'scoping protection policy' to the full revision period required. If you do see another paper with related content published elsewhere, nonetheless contact me immediately so that we can discuss the best way to proceed.

Please let us know if you prefer to discuss any of the revision recommendations directly, in particular if your laboratory suffers from COVID-19 related restrictions. We would be happy to discuss the individual revision points further with you.

To upload the revised version of your manuscript, please log in to your account: <https://lsa.msubmit.net/cgi-bin/main.plex>
You will be guided to complete the submission of your revised manuscript and to fill in all necessary information. Please get in touch in case you do not know or remember your login name.

Please note that papers are generally considered through only one revision cycle, so strong support from the referees on the revised version is needed for acceptance.

Thank you for this interesting contribution to Life Science Alliance. We are looking forward to receiving your revised manuscript.

Sincerely,

Reilly Lorenz
Editorial Office Life Science Alliance
Meyerohofstr. 1
69117 Heidelberg, Germany
t +49 6221 8891 414
e contact@life-science-alliance.org
www.life-science-alliance.org

B. MANUSCRIPT ORGANIZATION AND FORMATTING:

Reviewer #1 (Comments to the Authors (Required)):

Bhardwaj et al studied the E3-ligase and ATPase activities of 293T or HeLa cells that overexpressed normal RNF213 protein and mutant RNF213 proteins with E2488Q, I2999A and R4810K. They found that RNF213 uses UBE2D2 to catalyze K6-dependent poly-ubiquitination, and measured the E3-ligase activity by using this system. They found that the E3-ligase activities of mutant RNF213 proteins decreased, compared with normal RNF213. By contrast, the ATPase activity was not different between the normal and mutant RNF213 proteins. They concluded that RNF213 with R4810R mutation is a dominant negative mutant that decreases ubiquitylation globally.

Authors deal with critical issues how heterozygous RNF213 mutations increased the risk for MMD and the paper is well written.

- 1) There is a great concern regarding the assay system of E3-ligase activity of RNF213. Physiological E2 for RNF213 is not known and validation of use of UBE2D2 as E2 for the assay system remains unestablished.
- 2) All of the experiments were performed in the overexpressed RNF213 in cultured cells. Since overexpression experiments sometimes reflect unphysiological condition authors should provide the evidence of physiological conditions before they conclude that RNF213 with R4810R mutation works in a dominant-negative manner.

Reviewer #2 (Comments to the Authors (Required)):

The manuscript by Bhardwaj et al. discusses ubiquitination activities of the E3 ligase RNF213 that is mutated in Moyamoya disease. With its size of >5200 Aa, this is a huge protein, and it is commendable that the authors attempt to tackle a protein of such size, and indeed the cell biological data with full-length proteins are of very good quality. Moreover, expression suggests that the protein can be generated in good quantity and quality, which should be exciting for any cryo-EM person on campus.

Mutations in the RING domain cause expected loss of ubiquitination activity. Interestingly, also Moyamoya SNPs disrupt E3 activity (while not affecting ATPase activity). What is striking yet not discussed, is why/how an MMD mutant, R4810K, can affect RING E3 activity of a domain some 800 Aa upstream. A second striking finding is that The RNF213 RING domain seemingly directs the known, non-specific E2 ubiquitination activity of a widely used E2 enzyme, UBE2D2, towards assembling predominantly Lys6-linkages.

Clearly, there are interesting intrinsic mechanistic features in this exciting E3 ligase, which scream for a more detailed molecular/structural characterisation. Such analysis would be beyond the scope of the manuscript, which I enjoyed reading for its clarity and compelling findings, and which I would support to publish in Life Science Alliance once below minor comments are addressed.

- 1) There are now many complementary methods to gain insights into ubiquitin chain types, and the authors use a somewhat

outdated method, ubiquitin mutants. Especially concerning K6 chains, this has led to confusing data in the past (reason being, that K6 is also important for binding to and positioning Ub at the E2 enzyme). See work on MHC-I ubiquitination by Lehner and colleagues/Boname et al and also Goto et al. 2009/10.

Better approaches that would be highly doable for the in-vitro Fig 1 and potentially for the cell-based figures, are mass-spectrometry, and use of the K6-affimer (though this may be off the market at the moment), to provide additional evidence for K6 chain assembly.

2) E3 ligases are typically not specific for individual members of the UBE2D family. Fig 1c is cause for some concern. The authors should perform control experiments to show that their E1 preparation can charge all E2 enzymes similarly, and work with charged E2 enzymes. An additional control would be to use a different, well expressed E3 ligase with known non-discriminatory E2 usage for E2Ds - there are many GST-RING-members members that should work here, and which should be commercially available.

3) Why does the intensity of the Ub signal change with the use of the K-only mutants? (1e). Again, this data suggests that the other mutants are very differently charged by the E1 enzyme. I don't understand the strong 135kDa signal labelled as E1~Ub. I agree that it is suggestive of a difference in activity, but it is not clear why it should light up.

4) While challenging on a 5200 aa protein, some attempt at sec structure prediction should be performed. Is the region beyond the RING domain ordered/domain character and are any repeat sequences recognisable? What is known about other domains beyond those listed in Fig 1? What is the closest domain/protein similarity and species conservation for the region encompassing R4810K? this information would be valuable to add.

5) Methods lacks a section on molecular biology and sequencing, which given the size of the expressed protein, should be added. A Table with sequencing primers to dissect the 15kb insert should be included.

6) Figure labelling needs to be improved. Many labels are shifted, misaligned, or missing, and as such Figures cannot easily be understood. Colours in Figures are not easily distinguished (3 varieties of black in Fig 4c/d?)

Reviewer #3 (Comments to the Authors (Required)):

This manuscript describes an E3 ubiquitin ligase in which mutations have been strongly linked to Moyamoya disease (MMD). Here, the authors demonstrate that RNF213 functions in concert with the E2 UBE2D2 in the conjugation of -K6-linked ubiquitin chains and demonstrate that high penetrance variants of RNF213 result in a reduction of global protein ubiquitination, yielding insight into MMD. Findings from this work attempt to settle controversy on RNF213 ATPase activity, multimerization and ubiquitin function as causative activities for MMD. Overall, these findings are interesting and I have few comments related to necessary controls and to help settle controversial functions of RNF213.

Major Comments:

-The high molecular weight products in Figure 1c, 1d, and 1e must show that these modifications are indeed assembled on GST-RING RNF213 and not due to autoubiquitination of other enzymes present in the in vitro reaction such as E1 and E2s themselves. The authors must further show that the catalytic activity of GST-RING RNF213 is required for ubiquitination via UBE2D2. It is unclear how the ubiquitin chain mutants were generated in this assay as it is not clearly stated in the supplemental methods section. Further, the quantities/concentrations of all enzymes used in these reactions must be reported in the methods section.

-The authors' claim based on the data in figures 1d and 1e that other chain linkages can be formed on RNF213 based on the finding that adding back K6-linked chains only partially recovers poly-Ub of RNF213 to WT. It would be important for the authors to further explore this possibility through measurements of chain formation or adding back lysines (especially -K63) within ubiquitin in an attempt to recover levels to WT.

-Given the prior publication (Takeda_et_al, 2020) demonstrating Ubc13/Uev1a to have strong activity for RNF213 RING fragment and no activity with UbcH5b (UBE2D2), it will be critical for the authors to compare their findings in a side-by-side experiment using these two E2s and explain why the authors in this paper were not able to observe autoubiquitination activity in their assays.

-Given that full-length RNF213 can be expressed in cells, it is critical to establish that the full-length version of RNF213 can indeed assemble -K6 ubiquitin linkages.

-I disagree with the authors' conclusion that RNF213 ubiquitination is substantially reduced in UBE2D2 knockdown cells. The molecular weight of the HA-Ub reactive band is lower than the FLAG-RNF213 bands shown in the image (Figure 1h). How do the authors reconcile this discrepancy? And, if the assembly of RNF213 via UBE2D2 takes place via -K6 polyubiquitin linkages, why does this not appear as a higher molecular weight smear on the membrane, similar to Figure 3c?

-In addition to the coomassie gel stain blot, please provide a FLAG immunoblot of the immune purified proteins along with a vector only control with FLAG antibody lane.

-The ubiquitination experiment shown in Figure 3c should also be done in cells as opposed to a post-hoc ubiquitin assay.

-It is unclear how the experiments were conducted in Figure 3e. The amount of total protein ubiquitination is not a reflection of

the E3 ligase activity of RNF213 as one needs to show the ubiquitin is directly conjugated to RNF213. Thus, at minimum, the authors need to purify these proteins similar to what was done in Figure 3c or interpret their results based on global changes in ubiquitin patterning.

-Are the increases in Ub conjugates observed with RNF213 overexpression -K6 linked?

-The quantification of experiments from Figures 4a and b appear that the ATPase mutant (E2488Q) does not increase ubiquitination like the WT, suggesting that this activity is not serving in a dominant negative manner but is important to enhance protein ubiquitination. Moreover, it is unclear why the authors are not using the double mutant (E2488Q/E2845Q) that shows a lack of ATPase activity from Figure 2c.

Minor comments:

-In the introduction, state that protein ubiquitination selectively modifies lysine residues on a substrate.

-The authors discuss the common RNF213 R4810 mutation in the results section but never introduce this mutation in the prior text.

-There are other studies that assessed E3 ligase activity with one of the most common RNF213 variants. It is important to cite these key papers in the Introduction.

-Please provide original citations for the RNF213 reported patient mutations used in this study.

-There are a few grammatical errors in the text that need to be corrected.

Detailed Response to Reviewers' Comments

Reviewer #1 (Comments to the Authors (Required)):

<Summary>

Bhardwaj et al studied the E3-ligase and ATPase activities of 293T or HeLa cells that overexpressed normal RNF213 protein and mutant RNF213 proteins with E2488Q, I2999A and R4810K. They found that RNF213 uses UBE2D2 to catalyze K6-dependent poly-ubiquitination, and measured the E3-ligase activity by using this system. They found that the E3-ligase activities of mutant RNF213 proteins decreased, compared with normal RNF213. By contrast, the ATPase activity was not different between the normal and mutant RNF213 proteins. They concluded that RNF213 with R4810R mutation is a dominant negative mutant that decreases ubiquitylation globally.

<Comments>

Authors deal with critical issues how heterozygous RNF213 mutations increased the risk for MMD and the paper is well written.

1) There is a great concern regarding the assay system of E3-ligase activity of RNF213. Physiological E2 for RNF213 is not known and validation of use of UBE2D2 as E2 for the assay system remains unestablished.

We thank the Reviewer for this perceptive comment. While we were attempting to address his/her and the other Reviewers' concerns, it was reported that RNF213 has a novel E3 domain ("RZ"), distinct from the RING (Otten et al., 2021). Given these data, our initial findings using the isolated RING are at best incomplete and at worst potentially misleading. Consequently, we repeated all of the initial studies using purified full length (FL) RNF213. We also performed an extensive analysis of potential E2s.

Specifically, our new *in vitro* ubiquitylation and siRNA knockdown data suggest that UBE2D2 is an E2 for RNF213 *in vitro* and *in vivo* (Revised Figure 1d-g). Consistent with our results, an unbiased screen to find E2s for RNF213 showed that UBE2D2 can promote RNF213 autoubiquitylation, although it was not the only E2 with this activity (Ahel et al., 2020). Another unbiased screen indicated that RNF213 can interact with UBE2D2 as well as UBE2L3 (Pao et al., 2018).

That said, the aforementioned papers as well as our own new data (New Figure 1) indicate that other E2s (e.g., UBE2L3) also can act with RNF213, at least *in vitro*. Notably, however, we find that these E2s catalyze the addition of distinct UB linkages. We discuss this important issue in the revision.

2) All of the experiments were performed in the overexpressed RNF213 in cultured cells. Since overexpression experiments sometimes reflect unphysiological condition authors should provide the evidence of physiological conditions before they conclude that RNF213 with R4810R mutation works in a dominant-negative manner.

We agree conceptually with the Reviewer's concern regarding potential artifacts of over-expression. However, please note that WT RNF213 and various RNF213 mutants were re-expressed in RNF213 KO cells under doxycycline control, and the dox-induced RNF213 level is comparable to that of endogenous expression of RNF213 in these cells (e.g., Fig. 1C in the revision). Hence, we are not examining the effects of over-expression but rather, near-normal levels of these proteins.

Reviewer #2 (Comments to the Authors (Required)):

The manuscript by Bhardwaj et al. discusses ubiquitination activities of the E3 ligase RNF213 that is mutated in Moyamoya disease. With its size of >5200 Aa, this is a huge protein, and it is commendable that the authors attempt to tackle a protein of such size, and indeed the cell biological data with full-length proteins are of very good quality. Moreover, expression suggests that the protein can be generated in good quantity and quality, which should be exciting for any cryo-EM person on campus.

Mutations in the RING domain cause expected loss of ubiquitination activity. Interestingly, also Moyamoya SNPs disrupt E3 activity (while not affecting ATPase activity). What is striking yet not discussed, is why/how an MMD mutant, R4810K, can affect RING E3 activity of a domain some 800 Aa upstream. A second striking finding is that The RNF213 RING domain seemingly directs the known, non-specific E2 ubiquitination activity of a widely used E2 enzyme, UBE2D2, towards assembling predominantly Lys6-linkages.

Clearly, there are interesting intrinsic mechanistic features in this exciting E3 ligase, which scream for a more detailed molecular/structural characterisation. Such analysis would be beyond the scope of the manuscript, which I enjoyed reading for its clarity and compelling findings, and which I would support to publish in Life Science Alliance once below minor comments are addressed.

1) There are now many complementary methods to gain insights into ubiquitin chain types, and the authors use a somewhat outdated method, ubiquitin mutants. Especially concerning K6 chains, this has led to confusing data in the past (reason being, that K6 is also important for binding to and positioning Ub at the E2 enzyme). See work on MHC-I ubiquitination by Lehner and colleagues/Boname et al and also Goto et al. 2009/10.

Better approaches that would be highly doable for the in-vitro Fig 1 and potentially for the cell-based figures, are mass-spectrometry, and use of the K6-affimer (though this may be off the market at the moment), to provide additional evidence for K6 chain assembly.

We appreciate the Reviewer's criticisms, and we spent extensive time trying to perform the suggested experiments (indeed, much of the long time spent on this revision was aimed at addressing this comment). Also, as noted in our response to Reviewer 1, the experiments in the original submission analyzed the activity of the isolated RING domain and UBE2D2. In the revision, for the reasons discussed in our response to his/her comments, we repeated all of these experiments using FL RNF213 and UBE2D2 and UBE2L3.

Before submission of the initial manuscript, we had already attempted to use K6 affimers to assess RNF213 ubiquitylation. However, we found that this reagent (at least as provided by the company at the time) was unable to visualize either RNF213 or any other proteins (perhaps this is why this reagent is no longer on the market!). We also tried to identify RNF213 Ub sites by MS. Although we did not see K6-Ub in tryptic digests of immunopurified RNF213, we also did (could) not get full coverage of this large protein. To simplify the analysis, we also tried Lbpro (a ubiquitin clipase), but we found this enzyme be very inefficient.

Given these difficulties, we had no choice but to repeat the initial experiments using UB mutants. The results revealed clear, E2-dependent differences in the type of UB modification: UBE2D2 promoted primarily K6 ubiquitylation, whereas UBE2L3 catalyzed K48 and K11 ubiquitylation (and, to a much lower extent, K6 ubiquitylation). These results are found in Figures 1f and 1g of the revision. Although this approach might be "older", we believe that the results obtained are both reliable and highly interesting.

2) E3 ligases are typically not specific for individual members of the UBE2D family. Fig 1c is cause for some concern. The authors should perform control experiments to show that their E1 preparation can charge all E2 enzymes similarly, and work with charged E2 enzymes. An additional control would be to

use a different, well expressed E3 ligase with known non-discriminatory E2 usage for E2Ds - there are many GST-RING-members members that should work here, and which should be commercially available. **This issue is no longer relevant as, for the reasons detailed above and in response to Reviewer 1, we have removed all experiments using the GST-RING domain and only study purified FL RNF213.**

3) Why does the intensity of the Ub signal change with the use of the K-only mutants? (1e). Again, this data suggests that the other mutants are very differently charged by the E1 enzyme. I don't understand the strong 135kDa signal labelled as E1~Ub. I agree that it is suggestive of a difference in activity, but it is not clear why it should light up.

We appreciate this point as well, but it too is no longer relevant, as we have repeated these experiments with FL RNF213.

4) While challenging on a 5200 aa protein, some attempt at sec structure prediction should be performed. Is the region beyond the RING domain ordered/domain character and are any repeat sequences recognisable? What is known about other domains beyond those listed in Fig 1? What is the closest domain/protein similarity and species conservation for the region encompassing R4810K? this information would be valuable to add.

After the initial manuscript was submitted, the RNF213 structure was deciphered using cryoEM (Ahel et al., 2020)

5) Methods lacks a section on molecular biology and sequencing, which given the size of the expressed protein, should be added. A Table with sequencing primers to dissect the 15kb insert should be included.

We have included the requested information in the Methods section of the revised manuscript (under "Plasmids").

6) Figure labelling needs to be improved. Many labels are shifted, misaligned, or missing, and as such Figures cannot easily be understood. Colours in Figures are not easily distinguished (3 varieties of black in Fig 4c/d?)

We apologize for these errors, which have been corrected in the revised manuscript.

Reviewer #3 (Comments to the Authors (Required)):

This manuscript describes an E3 ubiquitin ligase in which mutations have been strongly linked to Moyamoya disease (MMD). Here, the authors demonstrate that RNF213 functions in concert with the E2 UBE2D2 in the conjugation of -K6-linked ubiquitin chains and demonstrate that high penetrance variants of RNF213 result in a reduction of global protein ubiquitination, yielding insight into MMD. Findings from this work attempt to settle controversy on RNF213 ATPase activity, multimerization and ubiquitin function as causative activities for MMD. Overall, these findings are interesting and I have few comments related

to necessary controls and to help settle controversial functions of RNF213.

Major Comments:

-The high molecular weight products in Figure 1c, 1d, and 1e must show that these modifications are indeed assembled on GST-RING RNF213 and not due to autoubiquitination of other enzymes present in the in vitro reaction such as E1 and E2s themselves. The authors must further show that the catalytic activity of GST-RING RNF213 is required for ubiquitination via UBE2D2. It is unclear how the ubiquitin chain mutants were generated in this assay as it is not clearly stated in the supplemental methods section. Further, the quantities/concentrations of all enzymes used in these reactions must be reported in the methods section.

As discussed in our responses to Reviewers 1 and 2, recent publications indicate that RNF213 has a novel E3 domain (RZ) (Otten et al., 2021), calling into question our initial experiments with GST-RNF213-RING. For this reason, we repeated all of the relevant experiments with full length RNF213, rendering the first part of this comment no longer relevant.

We apologize for not being clear in our Methods section. The requested information is present in the revision.

-The authors' claim based on the data in figures 1d and 1e that other chain linkages can be formed on RNF213 based on the finding that adding back K6-linked chains only partially recovers poly-Ub of RNF213 to WT. It would be important for the authors to further explore this possibility through measurements of chain formation or adding back lysines (especially -K63) within ubiquitin in an attempt to recover levels to WT.

We thank the Reviewer for this perceptive comment. As noted above, we have repeated the experiments with FL RNF213 and relevant E2s. Under these conditions, the relevant addback mutants alone or in combination have similar activity to the WT (Fig. 1g,h of the revision).

-Given the prior publication (Takeda et al., 2020) demonstrating Ubc13/Uev1a to have strong activity for RNF213 RING fragment and no activity with UbcH5b (UBE2D2), it will be critical for the authors to

compare their findings in a side-by-side experiment using these two E2s and explain why the authors in this paper were not able to observe autoubiquitination activity in their assays.

For the reasons discussed above, we no longer present any data on the GST-RING (because it might be confusing), but restrict our analysis to FL RNF213. We trust that the Reviewer will agree that our revised approach is more relevant than either the cited prior publication or our own initial data.

-Given that full-length RNF213 can be expressed in cells, it is critical to establish that the full-length version of RNF213 can indeed assemble -K6 ubiquitin linkages.

We thank the Reviewer for this comment, which motivated the aforementioned experiments on FL RNF213. As described above, our data suggest that UBE2D2-RNF213, and to a much lower extent, UBE2L3-RNF213, promote K6 ubiquitin linkage (Fig. 1f-g).

-I disagree with the authors' conclusion that RNF213 ubiquitination is substantially reduced in UBE2D2 knockdown cells. The molecular weight of the HA-Ub reactive band is lower than the FLAG-RNF213 bands shown in the image (Figure 1h). How do the authors reconcile this discrepancy? And, if the assembly of RNF213 via UBE2D2 takes place via -K6 polyubiquitin linkages, why does this not appear as a higher molecular weight smear on the membrane, similar to Figure 3c?

We apologize for the confusion, which was the result of inaccurate labeling of the original figure. We have repeated this result and provide the data in Figure 1d. In addition, as we used ECF (LICOR) with different colors for RNF213 and HA-UB, we provide the merged color blot, which shows the bands are the same. Also, please note that the "smeary" UB bands are from in vitro ubiquitylation experiments.

-In addition to the coomassie gel stain blot, please provide a FLAG immunoblot of the immune purified proteins along with a vector only control with FLAG antibody lane.

We have included the requested immunoblot in Fig. 2b of the revision.

-The ubiquitination experiment shown in Figure 3c should also be done in cells as opposed to a post-hoc ubiquitin assay.

We respectfully suggest that our new data with FL RNF213 clearly show that it has E2-dependent K6 Ub (as well as K11 and K48 Ub) activity.

-It is unclear how the experiments were conducted in Figure 3e. The amount of total protein ubiquitination is not a reflection of the E3 ligase activity of RNF213 as one needs to show the ubiquitin is directly conjugated to RNF213. Thus, at minimum, the authors need to purify these proteins similar to what was done in Figure 3c or interpret their results based on global changes in ubiquitin patterning.

We agree with the Reviewer's point regarding our interpretation of the results. Accordingly, we have changed the Results section to focus on global changes in ubiquitin patterning.

-Are the increases in Ub conjugates observed with RNF213 overexpression -K6 linked?

As stated in the manuscript (p. 12 in the revision), we reported previously (Banh et al, NCB 2016) that in cells with increased RNF213 activity, total K6-Ub was increased.

-The quantification of experiments from Figures 4a and b appear that the ATPase mutant (E2488Q) does not increase ubiquitination like the WT, suggesting that this activity is not serving in a dominant negative manner but is important to enhance protein ubiquitination. Moreover, it is unclear why the authors are not using the double mutant (E2488Q/E2845Q) that shows a lack of ATPase activity from Figure 2c.

We agree with the Reviewer's interpretation of the data. This finding is consistent with earlier work showing that this RNF213 mutant (E2488Q) does not oligomerize into the presumably active RNF213 hexamer (Morito et al., 2014). Therefore, one would expect the double mutant to be similarly inactive. This subtlety is clarified in the text.

Minor comments:

-In the introduction, state that protein ubiquitination selectively modifies lysine residues on a substrate.

We have modified the Introduction to state that "Mono-ubiquitylation joins a single UB molecule to a **lysine residue of the substrate, although multiple sites on the same substrate can be mono-ubiquitylated ("multi-mono-ubiquitylation"). Poly-ubiquitylation occurs when UB residues are added sequentially to a specific lysine residue in a previously conjugated UB, forming a UB chain."**

-The authors discuss the common RNF213 R4810 mutation in the results section but never introduce this mutation in the prior text.

Respectfully, this allele was mentioned on page 3 of the original version, where it also appears in the revision.

-There are other studies that assessed E3 ligase activity with one of the most common RNF213 variants. It is important to cite these key papers in the Introduction.

We are not clear on which papers the Reviewer is referring to, but would be happy to add these references if he/she provides.

-Please provide original citations for the RNF213 reported patient mutations used in this study.

Again respectfully, we did state in the initial (and do in the revised) Results that “Different RNF213 SNPs are associated with variable penetrance of MMD (Cecchi et al., 2014; Guey et al., 2017; Kamada et al., 2011; Liu et al., 2011; Sugihara et al., 2019).

-There are a few grammatical errors in the text that need to be corrected.

We have tried to ensure that all grammar is correct.

- Ahel, J., A. Lehner, A. Vogel, A. Schleiffer, A. Meinhart, D. Haselbach, and T. Clausen. 2020. Moyamoya disease factor RNF213 is a giant E3 ligase with a dynein-like core and a distinct ubiquitin-transfer mechanism. *Elife* 9:
- Cecchi, A.C., D. Guo, Z. Ren, K. Flynn, R.L. Santos-Cortez, S.M. Leal, G.T. Wang, E.S. Regalado, G.K. Steinberg, J. Shendure, M.J. Bamshad, G. University of Washington Center for Mendelian, J.C. Grotta, D.A. Nickerson, H. Pannu, and D.M. Milewicz. 2014. RNF213 rare variants in an ethnically diverse population with Moyamoya disease. *Stroke* 45:3200-3207.
- Guey, S., M. Kraemer, D. Herve, T. Ludwig, M. Kossorotoff, F. Bergametti, J.C. Schwitalla, S. Choi, L. Broseus, I. Callebaut, E. Genin, E. Tournier-Lasserre, and F. consortium. 2017. Rare RNF213 variants in the C-terminal region encompassing the RING-finger domain are associated with moyamoya angiopathy in Caucasians. *Eur J Hum Genet* 25:995-1003.
- Kamada, F., Y. Aoki, A. Narisawa, Y. Abe, S. Komatsuzaki, A. Kikuchi, J. Kanno, T. Niihori, M. Ono, N. Ishii, Y. Owada, M. Fujimura, Y. Mashimo, Y. Suzuki, A. Hata, S. Tsuchiya, T. Tominaga, Y. Matsubara, and S. Kure. 2011. A genome-wide association study identifies RNF213 as the first Moyamoya disease gene. *J Hum Genet* 56:34-40.
- Liu, W., D. Morito, S. Takashima, Y. Mineharu, H. Kobayashi, T. Hitomi, H. Hashikata, N. Matsuura, S. Yamazaki, A. Toyoda, K. Kikuta, Y. Takagi, K.H. Harada, A. Fujiyama, R. Herzig, B. Kirschek, L. Zou, J.E. Kim, M. Kitakaze, S. Miyamoto, K. Nagata, N. Hashimoto,

- and A. Koizumi. 2011. Identification of RNF213 as a susceptibility gene for moyamoya disease and its possible role in vascular development. *PLoS One* 6:e22542.
- Morito, D., K. Nishikawa, J. Hoseki, A. Kitamura, Y. Kotani, K. Kiso, M. Kinjo, Y. Fujiyoshi, and K. Nagata. 2014. Moyamoya disease-associated protein mysterin/RNF213 is a novel AAA+ ATPase, which dynamically changes its oligomeric state. *Sci Rep* 4:4442.
- Otten, E.G., E. Werner, A. Crespillo-Casado, K.B. Boyle, V. Dharamdasani, C. Pathe, B. Santhanam, and F. Randow. 2021. Ubiquitylation of lipopolysaccharide by RNF213 during bacterial infection. *Nature* 594:111-116.
- Pao, K.C., N.T. Wood, A. Knebel, K. Rafie, M. Stanley, P.D. Mabbitt, R. Sundaramoorthy, K. Hofmann, D.M.F. van Aalten, and S. Virdee. 2018. Activity-based E3 ligase profiling uncovers an E3 ligase with esterification activity. *Nature* 556:381-385.
- Sugihara, M., D. Morito, S. Ainuki, Y. Hirano, K. Ogino, A. Kitamura, H. Hirata, and K. Nagata. 2019. The AAA+ ATPase/ubiquitin ligase mysterin stabilizes cytoplasmic lipid droplets. *J Cell Biol* 218:949-960.
- Takeda, M., T. Tezuka, M. Kim, J. Choi, Y. Oichi, H. Kobayashi, K.H. Harada, T. Mizushima, S. Taketani, A. Koizumi, and S. Youssefian. 2020. Moyamoya disease patient mutations in the RING domain of RNF213 reduce its ubiquitin ligase activity and enhance NFkappaB activation and apoptosis in an AAA+ domain-dependent manner. *Biochem Biophys Res Commun* 525:668-674.

January 17, 2022

RE: Life Science Alliance Manuscript #LSA-2020-00807-TR

Dr. Benjamin G Neel
New York University Langone Medical Center
Medicine
522 1st Avenue
Smilow 12th Fl, Suite 1201
New York, New York 10016

Dear Dr. Neel,

Thank you for submitting your revised manuscript entitled "MMD-associated RNF213 SNPs Encode Dominant Negative Alleles That Globally Impair Ubiquitylation". We would be happy to publish your paper in Life Science Alliance pending final revisions necessary to meet our formatting guidelines.

- please address Reviewer 3's remaining minor comments
- please add the Twitter handle of your host institute/organization as well as your own or/and one of the authors in our system
- please note that titles in the system and manuscript file must match
- please consult our manuscript preparation guidelines <https://www.life-science-alliance.org/manuscript-prep> and make sure your manuscript sections are in the correct order
- please add your main and supplementary figure legends to the main manuscript text after the references section
- please use capital letters when introducing the panels in Figures, their legends, and callouts in the manuscript text
- please use the [10 author names, et al.] format in your references (i.e. limit the author names to the first 10)
- there is a callout for figure 3F in the manuscript text. Please revise
- please add a callout for Figure S2B to your main manuscript text

A. FINAL FILES:

B. MANUSCRIPT ORGANIZATION AND FORMATTING:

Sincerely,

Reviewer #2 (Comments to the Authors (Required)):

The authors have addressed my concerns and I'm happy to support publication of this interesting and topical manuscript.

Reviewer #3 (Comments to the Authors (Required)):

The authors provide a much improved version of the manuscript that is now well written that includes key experiments to resolve discrepancies in the literature surrounding RNF213. Thus, the authors have satisfied the majority of this reviewer's concerns. Some final minor comments:

It would be ideal for the authors to compare RNF213 activity with UBE2D2 to that of UBE2L3 within the same blot in Figure 3c to ensure that the decreased activity observed with UBE2L3 is not simply due to decreased exposure/contrast of the blot.

In the Results section title "MMD-associated SNPs encode mutants with decreased global ubiquitylation in cells", please make sure to cite the original paper that showed that the "E2488Q mutation prevents ATP hydrolysis, which is required for homo-hexamers disassembly".

The labels in the blots above Figure 1f do not appear to be well aligned with the representative immunoblots below.

Reviewer #2 (Comments to the Authors (Required)):

The authors have addressed my concerns and I'm happy to support publication of this interesting and topical manuscript.

We thank the reviewer for his/her support to publication.

Reviewer #3 (Comments to the Authors (Required)):

The authors provide a much improved version of the manuscript that is now well written that includes key experiments to resolve discrepancies in the literature surrounding RNF213. Thus, the authors have satisfied the majority of this reviewer's concerns. Some final minor comments:

It would be ideal for the authors to compare RNF213 activity with UBE2D2 to that of UBE2L3 within the same blot in Figure 3c to ensure that the decreased activity observed with UBE2L3 is not simply due to decreased exposure/contrast of the blot.

We thank the Reviewer for his/her valuable suggestions. In Fig 1f, we performed the requested autoubiquitination experiment using full length RNF213 with UBE2D2 or UBE2L3 in the same blot. The results clearly show that RNF213 autoubiquitination is much stronger with UBE2D2 compared to UBE2L3 (Fig 1f). Figure 3c was performed to address a different issue; i.e., the difference in autoubiquitination of RNF213 WT compared to MMD SNP mutant.

In the Results section title "MMD-associated SNPs encode mutants with decreased global ubiquitylation in cells", please make sure to cite the original paper that showed that the "E2488Q mutation prevents ATP hydrolysis, which is required for homo-hexamers disassembly".

We thank Reviewer for bringing this to our attention. We have changed the text and added the relevant reference to the sentence.

The labels in the blots above Figure 1f do not appear to be well aligned with the representative immunoblots below.

We again thank the Reviewer for bringing this to our attention. We have realigned the figure appropriately.

January 28, 2022

RE: Life Science Alliance Manuscript #LSA-2020-00807-TRR

Dr. Benjamin G Neel
New York University Langone Medical Center
Medicine
522 1st Avenue
Smilow 12th Fl, Suite 1201
New York, New York 10016

Dear Dr. Neel,

Thank you for submitting your Research Article entitled "MMD-Associated RNF213 SNPs Encode Dominant Negative Alleles That Globally Impair Ubiquitylation". It is a pleasure to let you know that your manuscript is now accepted for publication in Life Science Alliance. Congratulations on this interesting work.

DISTRIBUTION OF MATERIALS:

Again, congratulations on a very nice paper. I hope you found the review process to be constructive and are pleased with how the manuscript was handled editorially. We look forward to future exciting submissions from your lab.

Sincerely,
